# Potential Pharmacokinetic Effect of Chicken Xenobiotic Receptor Activator on Sulfadiazine: Involvement of P-glycoprotein Induction

**DOI:** 10.3390/antibiotics11081005

**Published:** 2022-07-26

**Authors:** Mei Li, Ziyong Xu, Wang Lu, Liping Wang, Yujuan Zhang

**Affiliations:** 1School of Biotechnology, Jiangsu University of Science and Technology, Zhenjiang 212100, China; 211211802106@stu.just.edu.cn (M.L.); 209310014@stu.just.edu.cn (Z.X.); 202211802226@stu.just.edu.cn (W.L.); 2MOE Joint International Research Laboratory of Animal Health and Food Safety, School of Veterinary Medicine, Nanjing Agricultural University, Nanjing 210095, China; wlp71@njau.edu.cn

**Keywords:** P-gp, induction, sulfadiazine, CXR, chickens

## Abstract

Studies on pharmacokinetic drug–drug interactions have highlighted the importance of P-glycoprotein (P-gp) because of its involvement in substrate drug transport. This study aimed to investigate the role of chicken xenobiotic receptor (CXR) in the regulation of P-gp and its influences on pharmacokinetics of P-gp substrate sulfadiazine. ALAS1 and CYP2C45, the prototypical target genes of CXR, were used as a positive indicator for CXR activation in this study. Results show that ABCB1 gene expression was upregulated, and transporter activity was increased when exposed to the CXR activator metyrapone. Using ectopic expression techniques and RNA interference to manipulate the cellular CXR status, we confirmed that ABCB1 gene regulation depends on CXR. In vivo experiments showed that metyrapone induced ABCB1 in the liver, kidney, duodenum, jejunum and ileum of chickens. In addition, metyrapone significantly changed the pharmacokinetic behavior of orally administered sulfadiazine, with a C_max_ (8.01 vs. 9.61 μg/mL, *p* < 0.05) and AUC_0-t_ (31.46 vs. 45.59 h·mg/L, *p* < 0.01), as well as a higher T_1/2λ_ (2.42 vs.1.67 h, *p* < 0.05), Cl/F (0.62 vs. 0.43 L/h/kg, *p* < 0.01) and Vz/F (2.16 vs.1.03 L/kg, *p* < 0.01). Together, our data suggest that CXR is involved in the regulation of P-gp, and, consequently, the CXR activator can affect, at least in part, the pharmacokinetic behavior of orally administered sulfadiazine.

## 1. Introduction

P-glycoprotein (P-gp, encoded by ABCB1), which belongs to the ATP-binding cassette (ABC) transporter family, is responsible for actively transporting different substrates from the intra- to the extracellular environment against their concentration gradients [1,2]. Sulfadiazine is an antimicrobial agent with bacteriostatic activity and is commonly used in poultry. It was shown that CYP2C9 is involved in sulfadiazine metabolism [3]. Our group previously demonstrated that sulfadiazine is actively transported by chicken P-gp but not breast cancer resistance protein (BCRP, another important member of the ABC transporter family) [4]. P-gp is expressed at high levels in pharmacologically important organs (e.g., intestine, kidney and liver), where it influences the pharmacokinetics and toxicity of substrate drugs [5,6]. Thus, the importance of researching the regulatory mechanisms of P-gp is becoming apparent.

The xenobiotic receptors pregnane X receptor (PXR) and constitutive androstane receptor (CAR) are two xenobiotic-sensing nuclear receptors and function as sensors of toxic byproducts, in order to enhance their elimination [7,8,9]. The regulatory mechanism for the above alterations is the binding of a chemical to PXR or CAR, followed by induction of enzymes or the ABC transporter family involved in drug metabolism [10,11,12]. A link between P-gp regulation and xenobiotic receptors has been established in rodents and humans [13,14]. However, no direct evidence for the regulation of chicken P-gp by chicken xenobiotic receptor (CXR) has been confirmed to date.

CXR was first cloned by the Christoph group and has a close relationship with PXR and CAR by sequence comparisons [15]. It is highly expressed in the main drug-metabolizing tissues and regulates xenobiotic-metabolizing enzymes such as CYP2C45 and CYP2H1 [16,17]. CXR heterodimerizes with the 9-cis-retinoic acid receptor (RXR, NR2B group) to bind to the DNA of CYP2C45 or CYP2H1, as evidenced in gel mobility shift assays. This binding occurs on a repeat of hexamer half-sites derived from the AG(G/T)TCA consensus sequence characteristic of all nuclear receptors. It was also noticed that CXR is responsible for the transcriptional activation of the ALAS1 gene by drugs. Electrophoretic mobility shift assays and transactivation studies demonstrate direct interactions between the nuclear receptor binding sites and CXR-implicating drug activation mechanisms for ALAS1 similar to those found in inducible CYP2C45 and CYP2H1 [18]. Here, we identified CXR as a positive regulator of chicken P-gp and, consequently, the CXR activator can affect, at least in part, the pharmacokinetic behavior of orally administered P-gp substrate sulfadiazine. Elucidation of this regulation has far-reaching significance for the development and usage of veterinary drugs.

## 2. Results

### 2.1. Metyrapone Upregulates Expression of ABCB1

To investigate whether CXR is involved in the regulation of ABCB1 expression, we first examined whether ABCB1 expression is modulated in chicken primary hepatocytes when exposed to the CXR agonist metyrapone [15,16,17,18]. ALAS1 and CYP2C45 are the prototypical target genes of CXR, as previously described, and, therefore, were used as a positive indicator of CXR activation. Compared to the controls, two different concentrations of metyrapone (100 and 500 μM) significantly up-regulated the mRNA levels of ALAS1 (1.87–2.61-fold, *p* < 0.05, *p* < 0.01), CYP2C45 (2–4.02-fold, *p* < 0.01) and ABCB1 (2.01–2.52-fold, *p* < 0.01) (Figure 1).

We further tested whether the observed induction of ABCB1 by metyrapone in chicken primary hepatocytes was primarily mediated by CXR using the CXR antagonist ketoconazole [19,20,21]. Cells were exposed to ketoconazole in the presence of metyrapone (500 μM) for 24 h. As expected, two different concentrations of ketoconazole (10 and 20 μM) prevented induction of ALAS1 (0.73–0.62-fold, *p* < 0.01), CYP2C45 (0.77–0.74-fold, *p* < 0.05) and ABCB1 (0.68–0.41-fold, *p* < 0.05, *p* < 0.01) by metyrapone, further suggesting that CXR is involved in the regulation of P-gp (Figure 2).

### 2.2. Agonist-Activated CXR Increases P-gp Function in Chicken Primary Hepatocytes

To determine whether induction of ABCB1 expression by metyrapone modulates the transport function of P-gp, Rho123 (a selective P-gp substrate) accumulation assay was performed using chicken primary hepatocytes (Figure 3). Intracellular Rho123 fluorescence was 21% lower in cells pretreated with 500 μM metyrapone for 24 h than in untreated cells (*p* < 0.01), indicating that P-gp-mediated efflux of Rho123 was increased. This metyrapone-induced efflux activity was reversed by co-treatment with the specific P-gp inhibitor verapamil (196% vs. control, *p* < 0.01). These results demonstrate that induction of ABCB1 expression by metyrapone is associated with an increase in the transport function of P-gp.

### 2.3. CXR Dependence of P-gp Induction

We performed gain-of-function assays by overexpressing CXR and loss-of-function assays by knocking down CXR in chicken primary hepatocytes. Transfection of chicken primary hepatocytes with the CXR expression vector significantly enhanced induction of ALAS1 (2.48-fold, *p* < 0.01), CYP2C45 (2.69-fold, *p* < 0.01) and ABCB1 (2.86-fold, *p* < 0.01) upon metyrapone treatment (Figure 4). By contrast, knockdown of CXR in chicken primary hepatocytes using siCXR (60% suppression of the CXR transcript level compared with NC siRNA treatment) attenuated induction of ALAS1 (0.54-fold, *p* < 0.01), CYP2C45(0.53-fold, *p* < 0.01) and ABCB1 (0.68-fold, *p* < 0.01) by metyrapone (Figure 5). These data suggest that activation of CXR is required for induction of chicken P-gp by metyrapone, indicating that CXR is directly involved in the regulation of P-gp.

### 2.4. CXR-Mediated Induction of P-gp in Chicken

The mRNA levels of ALAS1, CYP2C45 and ABCB1 were evaluated in the tissue of chickens after treatment with metyrapone. Compared to the controls, metyrapone treatment significantly increased expression of ALAS1 (3.26-fold in the liver, 1.63-fold in the kidney, 1.77-fold in the duodenum, 2.17-fold in the jejunum, 1.55-fold in the ileum), CYP2C45 (1.5-fold in the liver, 2-fold in the kidney, 3.79-fold in the duodenum, 2.04-fold in the jejunum, 2.13-fold in the ileum) and ABCB1 (3.35-fold in the liver, 3.67-fold in the kidney, 3.21-fold in the duodenum, 3.82-fold in the jejunum, 3.42-fold in the ileum) (Figure 6).

### 2.5. Agonist-Activated CXR Affects the Pharmacokinetics of the Orally Administered P-gp Substrate Sulfadiazine in Chickens

To assess whether induction of P-gp by metyrapone is functionally relevant, pharmacokinetic analysis of the P-gp substrate sulfadiazine was performed in chickens. The mean plasma concentration–time profiles of sulfadiazine orally administered alone or administered 24 h after CXR agonist metyrapone treatment are shown in Figure 7, and the relevant pharmacokinetic parameters are listed in Table 1. The combination of sulfadiazine/metyrapone significantly changed the pharmacokinetic behavior of orally administered sulfadiazine in chickens, with a lower C_max_ (8.01 vs. 9.61 μg/mL, *p* < 0.05) and AUC_0-t_ (31.46 vs. 45.59 h·mg/L, *p* < 0.01), as well as a higher T_1/2λ_ (2.42 vs.1.67 h, *p* < 0.05), Cl/F (0.62 vs. 0.43 L/h/kg, *p* < 0.01) and Vz/F (2.16 vs.1.03 L/kg, *p* < 0.01).

## 3. Discussion

P-glycoprotein (P-gp) is expressed in pharmacologically important tissues and transports a broad range of substrates against their concentration gradient [22]. P-gp protects the healthy body from foreign substances, but it has also become a major obstacle to disease treatment by restricting drug delivery to tissues. P-gp has attracted growing research efforts directed at its involvement in drug disposition (absorption, distribution and elimination) and DDIs [23]. Combined administration of drugs that inhibit or induce P-gp may increase or reduce systemic exposure to P-gp substrates, respectively [24]. An in vivo study with Arbor Acres broilers reported that enrofloxacin is more extensively absorbed upon coadministration of quercetin (a P-gp inhibitor) [25]. However, the bioavailability of orally administered enrofloxacin decreased from 72.5% to 24.8% by co-administration with rifampicin (a P-gp inducer) [26]. Therefore, it is of great importance to elucidate the molecular mechanisms that regulate P-gp expression.

The current study contributes to understanding P-gp regulation. Here, we found that CXR is a direct transcriptional regulator of chicken P-gp. It was shown previously that CXR was expressed in the main drug-metabolizing tissues (e.g., liver, kidney and small intestine) that affect the absorption, distribution, metabolism and excretion of drugs, and regulate genes encoding xenobiotic-metabolizing enzymes, such as chicken CYP2H1 and CYP2C45 [16,17]. This is the first report directly linking CXR to the regulation of an important ABC efflux transporter in chickens. Our findings expand the roles of CXR to adaptive regulation of the P-gp transporter.

We first examined the involvement of CXR in the regulation of P-gp using chicken primary hepatocytes, which are an in vitro model for elucidating the molecular mechanism underlying xenobiotic induction. ALAS1 and CYP2C45, the prototypical target genes of CXR, were used as a positive indicator of CXR activation. We found that the CXR activator metyrapone not only distinctly affected the expression of P-gp, but also improved the P-gp transporter activity in chicken primary hepatocytes. Unfortunately, we failed to detect the P-gp at protein level due to a lack of a suitable antibody. To exclude the possibility that other nuclear receptor pathways are involved in regulation of P-gp by metyrapone, RNA interference and ectopic expression techniques were used to manipulate the cellular CXR status. Induction of P-gp upon metyrapone treatment was significantly enhanced by overexpression of CXR and attenuated by silencing of CXR. These results confirm that metyrapone induces P-gp expression via a CXR-dependent mechanism. The CXR-mediated induction of P-gp in this study agrees with previous reports. In particular, Whyte-Allman et al. previously demonstrated at the blood–testis barrier that PXR and CAR regulate antiretroviral drug efflux transporters [27]. Manda et al. also showed the involvement of PXR in upregulating P-gp in human hepatic carcinoma cells after exposure to mitragyna speciosa and its alkaloids [28].

Consistent with the in vitro results, metyrapone induced expression of P-gp at the mRNA levels in the liver, kidney, duodenum, jejunum and ileum of chickens. The pharmacokinetic profiles of sulfadiazine, which has been shown to be the substrate of P-gp but not BCRP by our research [4], were changed when administration 24 h after metyrapone treatment. Substances other than P-gp may participate in the metabolic process of sulfadiazine in vivo. However, the results of this study suggest that modulation of the expression level and activity of P-gp by the CXR activator, at least in part, significantly change the systemic exposure of P-gp substrates. In the past few years, there has been sufficient evidence in rodents and humans that exogenous nuclear receptor induced P-gp in kidney, intestine and peripheral tissue can increase renal clearance, reduce drug bioavailability and reduce peripheral tissue distribution, respectively [29,30].

## 4. Materials and Methods

### 4.1. Reagents and Chemicals

Metyrapone, ketoconazole and Rho123 were purchased from Shanghai Yuanye Bio-Technology Co., Ltd. (Shanghai, China). Verapamil was purchased from MedChemExpress (Monmouth Junction, NJ, USA). DNA transfection reagents were purchased from Vazyme Biotech Co., Ltd. (Nanjing, China).

Real-time quantitative PCR primers were synthesized by Zhejiang Shangya Biotechnology Co., Ltd. (Zhejiang, China). Silenced CXR gene expression siRNA named siCXR was designed and synthesized by Sangon Bioengineering Shanghai (Stock) Co., Ltd. (Shanghai, China), and CXR eukaryotic expression plasmid PCDNA3.1-CXR was constructed in the laboratory before.

### 4.2. Isolation of Chicken Livers and Preparation of Primary Hepatocyte Cultures

Liver tissues were isolated from 14-day-old chicken embryos, digested by trypsin, centrifuged and sieved, and the primary liver cells were cultured in M199 medium supplemented with penicillin (100 IU/mL), streptomycin (100 μg/mL) and transferrin (5 μg/mL), according to a previously described method [31]. The cells were placed in a cell incubator at 37 ℃ 5% CO_2_ and 95% humidity.

### 4.3. RNA Isolation and RT-PCR Analysis

Total RNA was isolated using Trizol reagent (TAKARA, Dalian, China) from chicken primary hepatocytes and chicken tissue samples treated with different drugs. The concentration of RNA was determined using a microspectrophotometer, and the purity of RNA was checked by measuring the A260/A280 ratio. The RNA was reverse-transcribed into template cDNA using a reverse transcription kit (TAKARA, Dalian, China), and the chicken ABCB1, ALAS1 and CYP2C45 genes were detected using a real-time fluorescent quantitative PCR detection system. The chicken β-actin housekeeping gene was used as an internal control, and gene expression was analyzed by the 2^−∆∆ Ct^ method. All primer sequences are listed in Table 2.

### 4.4. Functional Detection of P-gp Activity

The primary hepatocytes of chicken embryo inoculated in 24-well plates were cultured to about 80% confluency. Cells were treated with metyrapone (500 μM) alone or combined with 100 μM verapamil, a specific P-gp inhibitor [32,33,34], for 24 h, and cells without treatment were used as the control. The cells were washed twice with PBS and then incubated with 5 μM Rho123, a selective P-gp substrate [35,36,37], for 60 min. The cells were digested by trypsin into a single-cell suspension and then washed with PBS 3 times. Rho123 fluorescence was detected by flow cytometry.

### 4.5. Overexpression of CXR

In this study, the CXR expression vector pcDNA3.1-CXR was constructed, and the specific operation was a CXR open reading frame of cloned chicken liver. Two enzyme digestion sites, EcoRI and XbaI, inserted a CXR open reading frame of cloned chicken liver into the pcDNA3.1(+) vector (Invitrogen, Carlsbad, CA, USA). Chicken primary hepatocytes were inoculated into 6-well plates and cultured to about 70% confluency. pcDNA3.1-CXR was transfected with Exfect 2000 (Vazyme Biotech Co., Ltd., Nanjing, China) and treated with Metyrapone (500 μM) for 24 h. Cells were collected to analyze the expression of ABCB1, ALAS1 and CYP2C45.

### 4.6. siRNA Studies

Chicken primary hepatocytes were transfected with CXR-targeting siRNA (siCXR) or negative control scrambled siRNA (NC siRNA) using Advanced DNA RNA Transfection Reagent (Zeta Life) for 24 h after being cultured to 70% confluency. Then, cells were treated with metyrapone (500 μM) for 24 h, and harvested to analyze ALAS1, CYP2C45 and P-gp expression. The siRNAs was synthesized from Sangon Biotechnology Co., Ltd. (Shanghai, China), and the sequences are listed in Table 1.

### 4.7. Experimental Animals and Sample Collection

HY-Line Brown commercial laying hens were purchased from a commercial poultry farm (Nanjing, Jiangsu, China). Chicken treatment procedures were approved by the Science and Technology Agency of Jiangsu Province (approval no. 2017–0007). Chickens at 200-days-old were randomly divided into two groups (6 in each group): one group was fed metyrapone (150 mg per chicken, orally) and the other group was not treated, as the control group. All chickens were fed a basic diet and rehydrated at the recommended humidity and temperature. Tissue samples were collected from the chickens 24 h after metyrapone feeding and were rapidly frozen in liquid nitrogen and stored at −70 °C until further analysis.

### 4.8. Pharmacokinetic Studies of Sulfadiazine in Chicken

A total of 12 200-day-old chickens were randomly divided into 2 groups. The first group was given sulfadiazine (20 mg/kg) by gavage. The second group was given metyrapone (150 mg/chicken) first, and sulfadiazine (20 mg/kg) orally after 24 h. At 0.17, 0.33, 0.5, 0.67, 0.83, 1, 2, 4, 6, 8, 10 and 12 h after sulfadiazine administration, the wing venous blood of each chicken was collected and placed in an anticoagulant tube containing heparin. The plasma was quickly collected by centrifugation at 5000× *g* for 5 min and stored at −80 °C until further analysis. Sulfadiazine concentration in plasma was determined by HPLC. Pharmacokinetic parameters were calculated using noncompartmental analysis and a computer program (WinNonlin 6.1, Phoenix Software, Los Angeles, CA, USA).

### 4.9. Statistical Analyses

Each experiment was performed at least three times. Comparison between groups were made using one-way analysis of variance (ANOVA) followed by Student’s paired *t*-test to determine the difference in significance with SPSS software (version 20.0, SPSS Inc., Chicago, IL, USA). Student’s *t*-test was used for studies comparing two independent groups. Data were plotted as means ± standard deviation. Differences were considered significant at *p* < 0.05, and *p* < 0.01 was considered an extremely significant difference.

## 5. Conclusions

In conclusion, our results demonstrate that CXR upregulates the P-gp/ABCB1 transporter, and the CXR activator metyrapone significantly changed, at least in part, the pharmacokinetic behavior of oral sulfadiazine. Therefore, xenobiotics may alter the pharmacokinetic properties of P-gp substrate drugs through nuclear receptor-mediated pathways if they are CXR agonists or antagonists. In the process of drug development, this may have far-reaching significance for evaluating the responsibility of DDI and determining appropriate DDI management strategies.

## Figures and Tables

**Figure 1 antibiotics-11-01005-f001:**
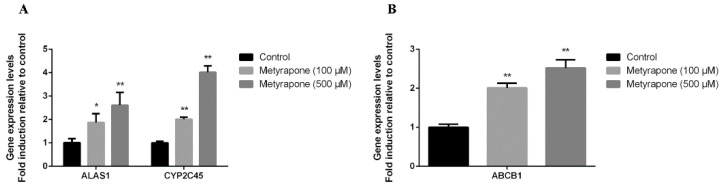
CXR-mediated upregulation of P-gp in primary hepatocyte cultures. Cells were exposed to various concentrations of the CXR agonist metyrapone for 24 h. RT-PCR analysis of ALAS1, CYP2C45 (**A**) and ABCB1 (**B**) mRNA levels. Bars show means ± SD of at least three independent experiments. * *p* < 0.05, compared with the control group; ** *p* < 0.01, compared with the control group.

**Figure 2 antibiotics-11-01005-f002:**
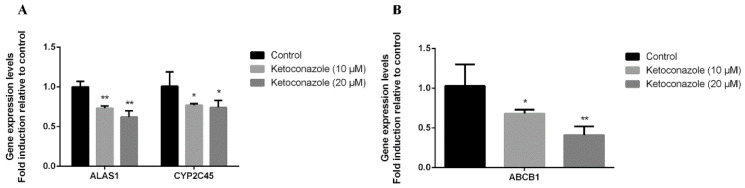
Effects of the CXR antagonist ketoconazole on P-gp expression in chicken primary hepatocytes. Cells were exposed to ketoconazole with or without the CXR agonist metyrapone for 24 h. RT-PCR analysis of ALAS1, CYP2C45 (**A**) and ABCB1 (**B**) mRNA levels. Bars show means ± SD of at least three independent experiments. * *p* < 0.05, compared with the control group; ** *p* < 0.01, compared with the control group.

**Figure 3 antibiotics-11-01005-f003:**
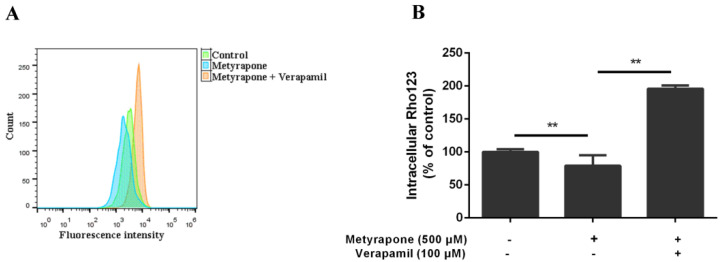
P-gp transport activity in chicken primary hepatocytes exposed to the CXR activator metyrapone. Intracellular Rho123 fluorescence was assessed in untreated cells and cells treated with 500 μM metyrapone for 24 h with or without pre-exposure to the P-gp-specific inhibitor verapamil (100 μM) for 1 h. (**A**)The histogram shows fluorescence (x axis) representing Rho123 accumulation plotted as a function of the number of cells (y axis). (**B**) Summaries of Rho123 accumulation. Bars show means ± SD of at least three independent experiments. ** *p* < 0.01.

**Figure 4 antibiotics-11-01005-f004:**
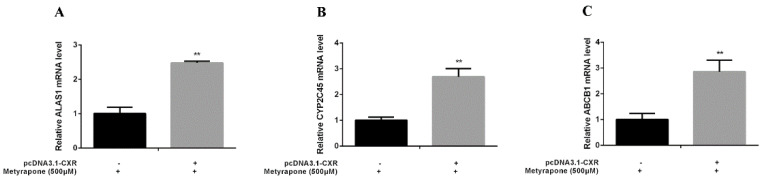
RT-PCR analysis of ALAS1 (**A**), CYP2C45 (**B**) and ABCB1(**C**) mRNA levels in chicken primary hepatocytes transfected with a CXR expression vector and then treated with 500 μM metyrapone for 24 h. Bars show means ± SD of three independent experiments. ** *p* < 0.01, compared with the control group.

**Figure 5 antibiotics-11-01005-f005:**
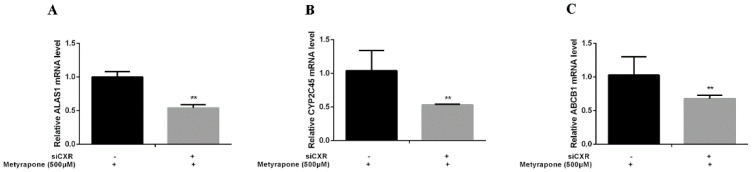
RT-PCR analysis of ALAS1 (**A**), CYP2C45 (**B**) and ABCB1(**C**) mRNA levels in chicken primary hepatocytes treated with 500 μM metyrapone after siRNA-mediated knockdown of CXR. Bars show means ± SD of three independent experiments. ** *p* < 0.01, compared with the control group.

**Figure 6 antibiotics-11-01005-f006:**
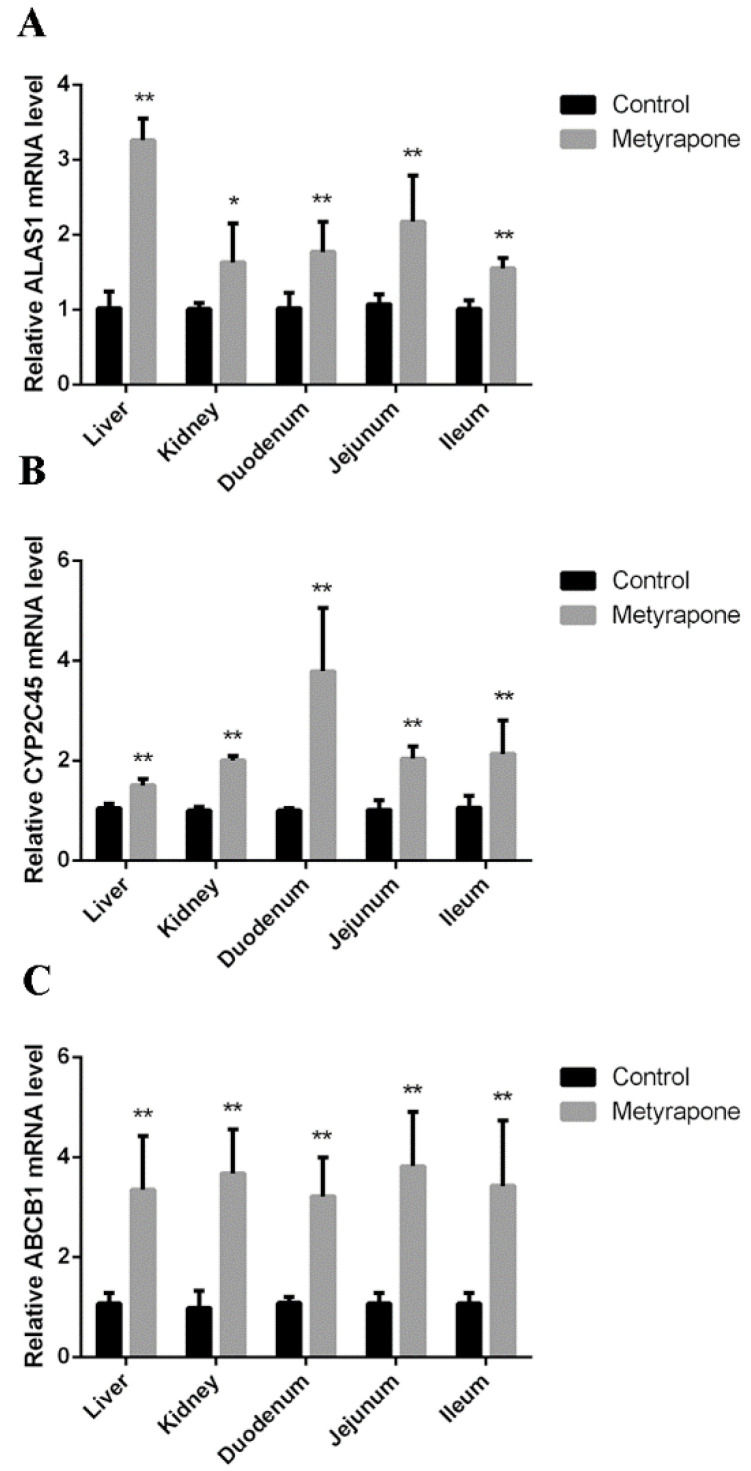
CXR-mediated induction of P-gp in chickens. RT-PCR analysis of ALAS1 (**A**), CYP2C45 (**B**) and ABCB1(**C**) mRNA levels with or without the CXR activator metyrapone. Bars show means ± SD of three independent experiments. * *p* < 0.05, compared with the control group; ** *p* < 0.01, compared with the control group.

**Figure 7 antibiotics-11-01005-f007:**
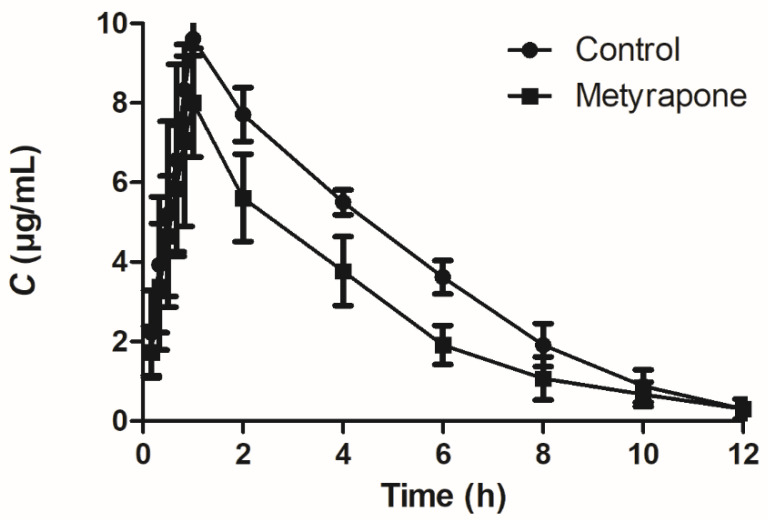
Plasma concentration–time profiles of sulfadiazine after single oral administration at a dose of 20 mg/kg body weight alone and administration 24 h after CXR agonist metyrapone treatment. Data represent means ± SD (*n* = 6).

**Table 1 antibiotics-11-01005-t001:** Pharmacokinetic parameters of orally administered sulfadiazine in chickens.

Parameter	Sulfadiazine	Sulfadiazine + Metyrapone	*p*-Value
C_max_ (μg/mL)	9.61 ± 0.41	8.01 ± 1.37 *	0.03
T_max_ (h)	0.94 ± 0.14	1.11 ± 0.44	0.40
T_1/2λ_ (h)	1.67 ± 0.38	2.42 ± 0.67 *	0.04
Vz/F (L/kg)	1.03 ± 0.19	2.16 ± 0.56 **	0.00
Cl/F (L/h/kg)	0.43 ± 0.02	0.62 ± 0.10 **	0.00
AUC_0-t_ (h·mg/L)	45.59 ± 1.87	31.46 ± 4.46 **	0.00

* *p* < 0.05; ** *p* < 0.01. Significant difference between parameters of sulfadiazine in the presence and absence of metyrapone in chickens. C_max_, the peak concentration; T_max_, time to reach peak concentration; T_1/2__λ_, the elimination half-life; Vz/F, apparent volume of distribution per fraction of the dose absorbed; Cl/F, plasma clearance per fraction of the dose absorbed; AUC_0-t_, area under the curve up to the last measurable concentration.

**Table 2 antibiotics-11-01005-t002:** Oligonucleotide sequences of primers.

Name	Primer Sequence (5′–3′)
Primers for plasmid construction	
pcDNA3.1-CXR-F	CCCAAGCTTATGTCCCAGTCCAGCCCCTCGGAC
pcDNA3.1-CXR-R	CTAGTCTAGATCAGCTGATGATTTCGGAGAGCAGCGGAGTCATGC
Primers for RNA interference	
siCXR-F	UCAGGCGCUCCAUCCUUAATT
siCXR-R	UUAAGGAUGGAGCGCCUGATT
NC-F	UUCUCCGAACGUGUCACGUTT
NC-R	ACGUGACACGUUCGGAGAATT
Primers for RT-PCR	
ABCB1 (P-gp)-F	ACCAGTCTCCCTATAGCAATG
ABCB1 (P-gp)-R	GGATATAAGCAGCCACAAGAAC
ALAS1-F	CATCTCTGGAACGCTCGGCAAG
ALAS1-R	CCAGCAGCATACGAACGGACAG
CYP2C45-F	CGATACGGGCTTCTGCTTCTTCTC
CYP2C45-R	TGGACCACTGCGTCTGTGTAGG
β-actin-F	TCCCTTCGGCATCCCTGTC
β-actin-R	GGCGTTGGTCTCCTCGTTG

## Data Availability

Not applicable.

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
