# Peer review of "Potential Pharmacokinetic Effect of Chicken Xenobiotic Receptor Activator on Sulfadiazine: Involvement of P-glycoprotein Induction"

_antibiotics, 2022, doi:10.3390/antibiotics11081005_

Round 1

Reviewer 1 Report

This paper aimed to report the regulation of P-gp in chicken through the chicken xenobiotic receptor (CXR). However, there are many unclear points and errors. Some specific points are detailed below.

1.     Abstract

-       Line 22-24; Units for pharmacokinetic parameters should be presented.

-       Line 21; The expression “Coadministration” is incorrect and may be confusing, because the metyrapone was administered 24 hours prior to sulfadiazine administration.

2.     Introduction

More specific evidence and references for the study design should be presented.

-       References for drug metabolizing enzymes and transporters responsible for sulfadiazine pharmacokinetics should be addressed.

-       The rationale for using metyrapone and ketoconazole as an activator and an antagonist of CXR, respectively, should be presented.

-       It is necessary to present the reported information on the regulation of drug metabolizing enzymes and transporters by CXR in more detail.

3.     Section 2.5.

-       Pharmacokinetic parameters (ex> T1/2λ, Vz) listed should be explained.

-       Line 142: unit for “Vz/F” parameter is wrong.

4.     Discussion

-       Line 158-159; I don’t agree with that this study is related to “clinical importance”.

-       Line 161-165; Proper references should be added.

-       Line 188-190; This point should be addressed with more caution. Metyrapone treatment may induce other drug metabolizing enzymes and cause the decreased systemic exposure of sulfadiazine. Therefore, literature or evidence on important drug metabolizing enzymes and transporters that determine sulfadiazine pharmacokinetics should be presented and considered in interpreting the results of this study.

5.     Typos and errors

-       Line 94, Line 249; “?” should be revised.

-       Line 265-266; “metiratone” may be “metyrapone”. Also, the dose “150 mg/mouse” should be revised to correct dose for chicken.

-       In addition to the above mentioned, there are many typos, so careful correction is required.

Reviewer 2 Report

The manuscript reports analysis of the role of chicken xenobiotic receptor (CXR) in regulation of P-gp and its influences on the P-gp-dependent transport of sulfadiazine, in chickens, in vitro and in vivo. The manuscript is well organized and well written, the graphs are informative.

A major comment:

The changes in the transport and pharmacokinetics of sulfadiazine may be mediated not only by the Pgp, but also by other efflux pumps. This is because: a) rhodamine 123 is apparently a substrate of several multidrug resistance-associated proteins, b) metyrapone induces pleiotropic pharmacological effects.

Please describe in a more detailed fashion the pharmacological effects of the compounds used in the study (metyrapone, ketoconazole, verapamil, Rho123), and their interaction with different transport systems (Pgp, BCRP, other pumps, …). Please revise accordingly the interpretation and the conclusions regarding the relationships of CXR activation or inhibition and the sulfadiazine PK.

Additional comments:

Line 144 (Figure 7 label) – please state that metyrapone was administered 24 hr before sulfadiazine.

Line 277 (statistical analyses) and the Figure labels – please state the type of the statistical test/s that were used to analyze each of the specific sets of data.

Line 34 – replace “agents” with “agent”

Line 94 - correct the units

Line 162 – replace “express” with “expressed”

Round 2

Reviewer 1 Report

- Please recheck the Cmax value of "sulfadizaine+metyrapone" group, because the value looks somewhat different from Fig. 7.

- Line 291: "metyratone" should be "metyrapone".

- Statistical analysis:  Student's t-test would be proper for studies comparing two independent groups.